# Decreased Expression of CD314 by NK Cells Correlates with Their Ability to Respond by Producing IFN-γ after BCG Moscow Vaccination and Is Associated with Distinct Early Immune Responses

**DOI:** 10.3390/vaccines11081297

**Published:** 2023-07-28

**Authors:** Adeliane Castro da Costa, Lília Cristina de Souza Barbosa, André Kipnis, Ana Paula Junqueira-Kipnis

**Affiliations:** 1Campus Goiânia, Goiás Estácio de Sá University, Goiânia 74063-010, ZC, Brazil; adeliane.castrodacosta@gmail.com; 2Department of Biosciences and Technology, Institute of Tropical Pathology and Public Health, Federal University of Goiás, Goiânia 74605-050, ZC, Brazil; lcristinasb19@ufg.br (L.C.d.S.B.); akipnis@ufg.br (A.K.)

**Keywords:** vaccine, innate immunity, heterogeneity response, tuberculosis, neutrophil, adaptive

## Abstract

The immune response to vaccines is complex and results in various outcomes. BCG vaccination induces innate and specific responses that can lead to protection against tuberculosis, and cross-protection against other infections. NK cells have been associated with BCG-induced protection. Therefore, we hypothesize that differences in NK cell status before BCG vaccination may have a role in the ability of BCG to activate the immune response. Participants of a clinical trial were evaluated after BCG vaccination. The participants were assigned to different groups according to variation in IFN-γ expression by NK cells between days 1 and 15 after BCG vaccination. Individuals that presented a higher increase in IFN-γ expression by NK cells presented reduced CD314 expression at day 1, and after vaccination an increase in inflammatory NK cells and CD4 T-cell expression of IL-17. A negative correlation between expression of CD314 at day 1 and that of IFN-γ by NK cells after BCG vaccination was observed. Participants with lower of IFN-γ expression by NK cells after BCG vaccination presented an increase in the cytotoxic NK subpopulation and CD4 T-cell expression of IL-17 and IFN-γ. In conclusion, the expression of CD314 by NK cells before BCG vaccination influences their IFN-γ responses, generation of NK subpopulations, and the specific T immune response at 15 days after vaccination.

## 1. Introduction

Bacillus Calmette–Guérin (BCG), developed 100 years ago from the live attenuated strain of *Mycobacterium bovis*, is widely administered to infants and children worldwide and constitutes the only effective form of protection against tuberculosis (TB) [1,2]. This vaccine induces an immune response, with effector cells of the Th1/Tc1 type producing IFN-γ, but CD8^+^ T cells comprise a smaller proportion than CD4^+^ T cells [3,4,5]. BCG vaccination induces a nonspecific innate immune memory response that protects against TB and unrelated diseases [6,7,8]. This response mainly requires monocytes/macrophages, neutrophils, dendritic cells and NK cells [2,7,8]. The hypothesis that BCG confers protection against unrelated diseases arose from the observation that vaccinated newborns and children had a lower rate of infection and mortality from viral and bacterial diseases [9,10,11]. Since then, several randomized controlled trials (RCTs) have aimed to analyze the potential of BCG vaccination in adults for activating the innate and adaptive immune responses. More recently, studies have demonstrated an important role for BCG in the induction and/or the activation of innate immune cells, such as NK cells, monocytes, and neutrophils [12,13,14].

NK cells are lymphoid cells originating from bone marrow hematopoietic stem cells [15]. These cells act in promoting the surveillance of the body’s own cells during infections and/or induction of the inflammatory response through the secretion of cytokines, such as IFN-γ [16]. NK cells constitute approximately 5–15% of peripheral blood mononuclear cells (PBMCs), and, based on the relative expression of CD56 and CD16 cell surface markers, they can be divided into distinct subsets [17,18,19,20]. Based on the density of cell surface markers, human CD56 NK cells can be termed CD56^bright^ or CD56^dim^, which have different phenotypic properties [2]; that is, cytotoxic (CD56^dim^CD16^+^) or proinflammatory (CD56^bright^CD16^−^).

Although these cells have been reported to be important for protection against fungal and viral infections through the production of proinflammatory cytokines such as IFN-γ [13,21], it has been demonstrated that their response to vaccines can present in several variant forms, probably due to the genetic diversity of the population [22,23,24].

NK cells from the peripheral blood of healthy volunteers were shown to become activated and secrete higher levels of proinflammatory cytokines than NK cells from unvaccinated volunteers after BCG vaccination; this response was maintained for at least four months [13,25]. It is also known that BCG vaccination can promote myelopoiesis, favoring an increase in the number of neutrophils caused by the activation of genes linked to the transcription of human hematopoietic stem cells (HSPCs) [26]. The absence of neutrophils also influences the hyporeactivity of NK cells, a condition in which these cells produce less IFN-γ [27]. In a study by Moorlag et al. [28], it was demonstrated that, after BCG vaccination, there was an increase in the number of mature neutrophils in the peripheral circulation [28]. Although several clinical studies have evaluated the innate immune response induced by BCG, to our knowledge, no work has correlated NK cell activation with neutrophil activation after BCG vaccination.

In an RCT conducted by our research group [29,30], it was verified that the BCG Moscow vaccine, although safe, was not effective in inducing protection against COVID-19. Although NK cells producing IFN-γ and TNF-α were induced in the blood of the volunteers vaccinated with BCG, no differences were observed compared to NK cells from the nonvaccinated group. Additionally, participants who had a greater increase in IFN-γ^+^ NK cells had fewer incidences of symptoms such as fever and dyspnea when they presented with COVID-19 infection, independent of the BCG vaccination [30]. These results may indicate that activated NK cells that produce IFN-γ are important for regulating the immune response elicited by an infection or a vaccination. As shown by Angelo et al. [31], NK cells from the blood of healthy donors present great variability in their receptors, which may affect the ability of NK cells to respond to vaccination as well as to cooperate in immune responses [31]. Therefore, we hypothesize that differences in the NK cell status or phenotype before BCG vaccination may have a role in BCG vaccine responses.

This study aimed to evaluate whether the initial blood NK cell phenotype is related to the ability of BCG to activate the immune response.

## 2. Materials and Methods

### 2.1. Trial Design

The individuals included in this study participated in a unicentric parallel, randomized, phase II clinical trial conducted with health workers (HWs) with no prior COVID-19 infection. The study was conducted at the Federal University of Goias (UFG), Brazil, between 5 August 2020 and 31 August 2021, according to the protocol and ethical statements published by Junqueira-Kipnis et al. [29].

### 2.2. Protocol Registration

The clinical trial was registered on 5 August 2020 at REBEC (Registro Brasileiro de Ensaios Clínicos), Register number: RBR-4kjqtg and at WHO with trial registration number UTN: U1111-1256-3892 and approved by the Brazilian Ethics Committee with CAAE identification number: 31783720.0.0000.5078.

### 2.3. Innate Immune Response after BCG Vaccination

#### 2.3.1. NK and Neutrophil Evaluation

Heparinized total blood was drawn from the vaccinated participants on the day of inclusion (day 1) and 15 days after randomization. Briefly, after participant clinical trial inclusion, the blood was drawn, and the participant received a sealed envelope containing their group assignment, “vaccination” or “no vaccination” with BCG [29]. Individuals that were randomly assigned to the “vaccination” group were intradermally vaccinated with 100 μL (10^6^ CFU) of the BCG Moscow vaccine in the deltoid region of the right arm by an experienced nurse. In this study, 60 individuals who received the BCG Moscow vaccine were evaluated. For the comparison of NK cells producing IFN-γ at days 1 and 15, 48 individuals from the “no vaccination” group were also analyzed. The median age of the participants was 42 years old, ranging from 18 to 60 years old; most of them were female (68%). All participants had previous BCG scars and were exposed to several infectious diseases including tuberculosis, once they worked at the General Hospital.

Five hundred microliters of blood were aliquoted in cryogenic tubes containing 500 µL of freezing solution (20% DMSO and 80% bovine serum albumin [BSA]). All blood samples were kept at −80 °C until their cell preparations were ready for flow cytometry analysis.

To evaluate NK cell activation, the methodology of Anjos et al. [30] was adopted. Briefly, cells were incubated with 0.5 µg/µL BCG culture filtrate protein (CFP) for 2 h at 37 °C in a 5% CO_2_ incubator. The NK subpopulations were evaluated according to the intensity of fluorescence of CD56 and CD16 among gated CD3-negative lymphocytes. All antibodies used in this study were from eBioscience, and the clones of all antibodies are presented in the Appendix A.

To evaluate neutrophils, 200 μL/well of thawed blood suspension was distributed in 96-well plates followed by incubation for 10 min with 20 μL of mouse sera. After incubation, the erythrocytes were lysed with lysis solution (0.15 M NH_4_Cl, 10 mM HCO^3^), and the cells were washed twice with cRPMI. Then, the cells were incubated with the following mouse anti-human fluorescent antibodies: CD16-FITC; CD49d-PE; CD63-PercP; CD15-PE-Cyanine7; CD66-APC; CD14-Alexa Fluor 700; CD123-efluor 780 and CD16-APC e-Fluor 780 for 30 min at 4 °C in the dark. Then, the cells were incubated with Permfix and Permwash (BD Biosciences, San Jose, CA, EUA) solution. After incubation, the plate was centrifuged, and the supernatant was discarded and incubated with mouse anti-human antibody against Arginase-Alexa Fluor 700 or TNF-α-PE for 30 min at 4 °C in the dark. After washing with Permwash, the cells were resuspended in 500 μL of PBS with 0.05% sodium azide.

#### 2.3.2. Lymphocyte Evaluation

PBMCs were isolated from heparinized blood venous samples by density sedimentation with Ficoll-Paque Plus according to the manufacturer’s instructions. All PBMC samples were kept at −80 °C until cell preparation for cytometry. For flow cytometry, PBMCs were quickly thawed under agitation at 37 °C, and 200 μL of PBMCs were transferred to 96-well plates. The cells were stimulated with 1 μg/mL of recombinant antigen Ag85A and incubated with costimulatory antibodies (CD28/CD49d) BD FastImmune for 90 min at 37 °C/5% CO_2_ or incubated with media alone (negative control) or phytohemaglutinin (PHA) as a positive control. After incubation, the cells were treated with monensin 1% *v*/*v*—Golgi Stop Solution (3 mM; eBioscience; San Jose, CA, USA) and incubated for 5 h. After incubation, the plates were centrifuged at 2000 rpm for 20 min/10 °C, and the cells were resuspended and treated with 10 μL of mouse serum. After 10 min of incubation at 4–8 °C, surface staining with CD4-FITC was performed, followed by intracellular staining using a combination of the following antibodies: IL-22-PE-CY7, IL-2-APC, IFN-γ-eFluor 700, and IL-17-APC eFluor 780 (Appendix A). Briefly, first, the cells were incubated for 30 min with 25 μL of surface antibodies, treated with Permfix solution for 20 min/4 °C and permeabilized with Permwash solution (BD Biosciences, San Jose, CA, EUA) for 20 min/4 °C in the dark. Afterwards, the plates were centrifuged at 2000 rpm for 20 min at 10 °C, and 25 μL of intracellular antibodies were added per well. After washing with Permwash, the cells were resuspended in 200 μL of PBS containing 0.05% sodium azide.

#### 2.3.3. Flow Cytometry

The cells were immediately sorted using an Attune™ NxT flow cytometer (Applied Biosystems LifeTech—ThermoFisher, Waltham, MA, USA). To evaluate NK cells, at least 30,000 events were acquired and analyzed; for neutrophils and lymphocytes, 200,000 events were acquired and evaluated. The data were evaluated using FlowJo software version 10.1. All cells were gated to exclude doublets using FCS-A and FCS-H parameters. After this, based on granularity and size, lymphocytes were gated, and CD16^+^CD56^+^ cells were selected. These cells were analyzed for CD3 expression, and CD16^+^CD56^+^ CD3^−^ cells were considered to be NK cells. The median fluorescence intensity (MFI) of markers CD56, CD57, CD27 and CD314 was evaluated. For CD4^+^ lymphocytes, cells were evaluated for the cytokines described earlier. In neutrophil evaluation, size and granularity were assessed (FSC and SSC), and CD15^+^CD16^+^ cells were considered neutrophils. The intensity of fluorescence of arginase, TNF-α or surface molecules was evaluated. The gating strategy is presented in Appendix A.

### 2.4. Statistical Analysis

All data generated by FlowJo analysis were compiled in an Excel file and transferred to GraphPad Prism 9 software (Boston, MA, USA) to generate graphs and perform statistical analysis.

To determine the ranking criteria for participants according to the IFN-γ response to BCG vaccination, the NK cell IFN-γ^+^ percentage of all participants was classified according to the response magnitude at days 1 and 15 postvaccination. Then, a tricolour scale starting with yellow, light blue for the median point at a percentage of 10, and dark blue for the highest value was used to create a heatmap using Excel for both time points. The variation between day 15 and day 1 for each participant was used to group individuals.

All data were evaluated for normality using the Shapiro–Wilk normality test. Kruskal–Wallis and Dunn’s multiple comparison analyses were used for nonparametric data from different groups of participants. The correlation between two pairs of data was evaluated using a one-tailed Pearson correlation test. For T-cell-specific responses to Ag85, the comparison between day 1 and day 15 was performed using the Wilcoxon test. Statistically significant *p* values were as follows: * < 0.05; ** < 0.01; *** < 0.001; and **** < 0.0001.

## 3. Results

### 3.1. Heterogeneity of IFN-γ-Producing NK Cell Responses after Vaccination with BCG Moscow

To evaluate whether the NK phenotype before vaccination with BCG is important for the IFN-γ response after immunization, IFN-γ^+^ NK cells were classified according to the response magnitude and variation. As shown in Figure 1, NK cells from individuals vaccinated with BCG Moscow had different responses regarding IFN-γ positivity after immunization that could be arranged into three distinct groups. There were individuals who had NK cells with IFN-γ production before vaccination that diminished after immunization (group 1); individuals whose NK cells produced more IFN-γ after vaccination (increased NK cells upon vaccination, group 2); and individuals whose NK cell-IFN-γ production was maintained after BCG vaccination (group 3). Conversely, though NK cells from nonvaccinated individuals presented great variability in IFN-γ production at day 1, they did not present the same variation pattern as vaccinated individuals when compared to their status on day 15 (Appendix A). Thus, these results may indicate that the baseline NK cell characteristics could have influenced the NK cell response after BCG vaccination.

To assess the initial NK cell phenotype, the MFI of CD314 (NKG2D), CD27, CD57 and CD56 on NK cells at baseline was evaluated considering the above-defined groups (Figure 2). Individuals with NK cells with higher IFN-γ production in response to the BCG Moscow vaccine (group 2) presented NK cells with lower expression of CD314 MFI at baseline when compared with group 1 (Figure 2A; * *p* < 0.05). To evaluate whether the NK surface phenotype would change differently upon BCG vaccination, NK cells were also evaluated after immunization. Fifteen days after BCG vaccination, it was observed that individuals who presented NK cells with higher IFN-γ production upon BCG vaccination (group 2) continued to have lower CD314, and the CD56 expression of NK cells was lower than that of individuals in groups 1 and 3 (Figure 2E; * *p* < 0.05). The CD314 expression of NK cells at 15 days postvaccination decreased compared to day 1 among individuals in group 2, and no variation was observed for the other two groups (data not shown).

Correlation analysis revealed a negative correlation between the magnitude of IFN-γ response in NK cells post-BCG vaccination and initial expression of CD314 (Figure 3). These results could indicate that higher expression of CD314, an activating NK receptor, before vaccination may contribute to lower production of IFN-γ upon BCG vaccination. No correlation was observed for the other evaluated molecules (data not shown).

The reduction in CD56 expression by NK cells from individuals in group 2 after BCG vaccination compared to groups 1 and 3 may have impacted the NK subpopulations and consequently their function. Four NK subpopulations, according to Poli et al. [32] and Amand et al. [33], were evaluated: CD56^bright^CD16^−^, CD56^bright^CD16^+^, CD56^dim^CD16^−^ and CD56^dim^CD16^+^, and differences were verified only in CD56^bright^CD16^+^ NK cells and CD56^dim^CD16^−^ NK cells. NK cells from vaccinated individuals who presented a higher magnitude of IFN-γ response (group 2) presented a higher frequency of CD56^bright^CD16^+^ NK cells (Figure 4A gate 1) after immunization than the other groups (Figure 4D, * *p* < 0.05). In contrast, upon BCG vaccination, groups 1 and 3 showed an increase in the percentage of the CD56^dim^CD16^−^ NK cell subpopulation (Figure 4A gate 2; Figure 4C,G, * *p* < 0.05).

To evaluate which NK subpopulation contributed to the IFN-γ response after BCG vaccination, the percentage of IFN-γ^+^ NK cells was quantified (Figure 5). The percentage of NK CD56^bright^CD16^+^, CD56^bright^CD16^−^ and CD56^dim^CD16^+^ subpopulations positive for IFN-γ was increased only in group 2. Possibly, these NK subpopulations contributed to the overall BCG IFN-γ response 15 days postvaccination. Because it was observed that before vaccination the expression of CD314 was negatively associated with the IFN-γ response to BCG, the CD314 expression was also evaluated for each NK subpopulation. NK CD56^bright^CD16^−^ and CD56^bright^CD16^+^ cells presented similar MFI of CD314 among all groups, while NK CD56^dim^CD16^−^ and CD56^dim^CD16^+^ cells from group 2 presented reduced CD314 expression compared to the other groups.

### 3.2. Similar Neutrophil Surface Phenotypes but Different Responses to BCG Vaccination according to the Magnitude of NK Cell IFN-γ Response

Since the NK subpopulations responded differently according to their initial phenotype, we questioned whether neutrophils could also be different at baseline. First, no differences were observed while evaluating the frequency of neutrophils before or after BCG vaccination (frequency of neutrophils before vaccination: 54.31 ± 11.26 at day 1 and 56.33 ± 9.79 after 15 days of immunization, *p* = 0.35). Although no differences among the activating surface markers were observed between the groups on day 1, 15 days post-BCG vaccination, neutrophils from group 2 showed reduced expression of CD49d, CD63, CD66 and CD123 compared to group 1 (Figure 6). Analyzing some of the neutrophil intracellular molecules, it was observed that the production/positiveness of TNF-α was not modified by BCG vaccination and was not different among the participants that were classified according to the magnitude of NK IFN-γ responses (Figure 7B,D). Additionally, no intragroup differences were observed before and after vaccination (data not shown). Regarding arginase expression, neutrophils from participants in groups 1 and 3 presented higher expression at day 1 than neutrophils from participants in group 2. Neutrophils from participants in group 2 retained lower levels of arginase than neutrophils from the other two groups 15 days postvaccination (Figure 7A,C).

### 3.3. Vaccination with BCG Moscow Induces Specific T-Cell Responses but Heterogenic Phenotypes according to NK IFN-γ Responses

The heterogeneity of the NK and neutrophil responses after BCG vaccination may favor different T-cell responses. To evaluate this, the global T-cell-specific response among vaccinated participants was first evaluated. As shown in Figure 8, BCG Moscow vaccination induced T-cell-specific immune responses. CD4 T-cells from vaccinated subjects were able to produce IFN-γ, IL-17, IL-22 and IL-2 upon *Mycobacterium tuberculosis* recombinant Ag85 in vitro stimulation. These results confirm that the BCG Moscow vaccine was able to activate specific immune responses. However, the T-cell responses from vaccinated participants were not homogeneous regarding the different cytokines evaluated. Thus, we questioned whether there were differences in the CD4 T-cell-specific phenotypes based on NK-IFN-γ responses to BCG.

Interestingly, individuals who had reduced NK-IFN-γ^+^ percentage upon BCG vaccination (group 1) were able to induce CD4^+^ T-cell responses for all cytokines evaluated (IFN-γ, IL-17 and IL-22, Figure 9A, Figure 9B and Figure 9C, respectively), contrary to participants from groups 2 (increased NK-IFN-γ responders) and 3 (without NK-IFN-γ responses) that did not induce specific IFN-γ responses to Ag85. Participants from group 2 also did not induce CD4^+^ T-cell specific IL-22 responses. All groups were able to induce specific IL-17 responses to Ag85. Only groups 2 and 3 induced CD4+ T-cell-specific IL-2 (data not shown). Because the main differences among these groups were that the initial NK responses to BCG and NK-IFN-γ responses were negatively correlated with CD314 expression, we questioned whether this phenomenon could also be associated with the induction of IFN-γ by specific CD4 T-cells, but no correlation between NK CD314 basal expression and CD4 T-cells expressing IFN-γ at day 15 was observed (Appendix A).

## 4. Discussion

Immune responses in humans are heterogeneous and influenced by many external factors, such as infections and vaccines, as well as genetics, sex and pathological conditions [22]. Additionally, Esin and Batoni [34] suggested that variability in NK cell responses to BCG vaccination may be related to the degree of sensitization to mycobacterial antigens due to prior exposure to vaccination with BCG or environmental mycobacteria. Such factors may influence the immune response induced by vaccines, and at 15 days after vaccination with BCG Moscow, vaccinated individuals presented mixed levels of IFN-γ production by NK cells (Figure 1). Considering this heterogeneity in the immune response, some individuals vaccinated with BCG Moscow were classified as having the capacity for IFN-γ production by NK cells. There were individuals who had NK cells with decreased IFN-γ production at day 15 compared to baseline (group 1), individuals who produced more IFN-γ after vaccination (group 2), and individuals whose IFN-γ production did not differ between days 1 and 15 (group 3).

Human NK cells express a large set of receptors that, after interaction with specific cellular ligands [35], trigger a complex network of signals, which is regulated by the balance between the expression of activating and inhibitory molecules [20,34,36,37]. The combination of these molecules promotes additive, synergistic or redundant actions in NK cells [38], which individually or together can interfere with the degranulation of NK cells and cytokine production [39]. Direct interaction between human NK cells and BCG surface components promotes activation [34], proliferation and IFN-γ expression by these cells [40].

Group 1 presented reduced production of IFN-γ by NK cells 15 days after vaccination (Figure 1). Such NK cells had increased CD314 expression compared to group 2 at baseline (Figure 2A). In addition, both the CD314 and the CD56 molecules were increased 15 days after vaccination (Figure 2E,G) compared to the other groups. Interestingly, NK cells from individuals in group 3 that did not change their ability to produce IFN-γ upon BCG vaccination and in vitro stimulation also presented higher levels of CD314 and CD56 than NK cells from individuals in group 2. Our results suggest that it is possible that high levels of CD314 expression may inhibit their ability to produce IFN-γ after vaccination and in vitro stimulation with BCG cultured filtrate proteins (BCG-CFP). Conversely, low levels of CD314 expression by NK cells may result in a higher capacity to produce IFN-γ after vaccination and in vitro stimulation with CFP. The in vitro stimulation with BCG-CFP showed that after BCG vaccination, CD56^dim^CD16^+^ and CD56^dim^CD16^−^ NK subpopulations (Figure 4A and Figure 5C,D) from group 2 subjects presented reduced levels of CD314; thus, they were the main contributors for the reduced overall NK CD314 expression (Figure 2E) compared to the other groups. The ensuing question would be that low expression of CD314 by NK subpopulations would be associated with a high positivity for IFN-γ. That was not the case, as the NK subpopulations CD56^bright^CD16^+^ and CD56^bright^CD16^−^ were the main groups positive for IFN-γ and at 15 days post vaccination, they did not present differences in CD314 expression when compared amid the different groups. Therefore, the lower expression of CD314 is probably a phenotype associated with the initial status that may be directly or indirectly related to the IFN-γ responses to BCG, but it is not a phenotype associated with higher expression of IFN-γ and further studies should be conducted to understand the mechanisms involved in this characteristic.

The expression of CD314 and CD56 seems to contribute to the induction of a cytotoxic NK cell profile (CD56^dim^CD16^−^; Figure 4C). The interaction of CD314 with its ligand has been shown to favor the activation of STAT5, which is important for producing perforins, granzymes and NK cell cytotoxic activity [41,42], and to favor the activation of the transcription factor DAP, culminating in the activation of ERK [43,44], which promotes the generation of proinflammatory cytokines such as IL-6 and IL-1β [45]. Thus, it is possible to propose that mature CD56^dim^CD16^−^ NK cells could explain the activation of neutrophils and generation of Th17 observed here.

On the other hand, group 2 participants showed an increase in inflammatory CD56^bright^CD16^+^ NK cells after BCG vaccination. According to our results, this group had NK cells with lower expression of CD314 before vaccination, which was even more reduced after BCG vaccination. Although controversial, the process of NK cell subpopulation maturation is thought to transition between the CD56^bright^ stage, which is associated with an inflammatory response (TNF-α; IFN-γ, TNF-β producers), and the CD56^dim^ stage, which comprises a cytotoxic response [46,47,48]. The expression of human CD314 on the surface of NK cells is directly related to signal transducer and activator of transcription 3 [49]. The reduction in CD314 expression by NK cells 15 days after vaccination may be a result of low activation of STAT3, which in turn may be due to the low basal expression of CD314 in this group. This could corroborate the finding that individuals from this group did not show an increase in IL-22-producing CD4^+^ T cells, whose generation is also dependent on STAT3 [50]. To produce IFN-γ, NK cells need to activate STAT-1, which is stimulated by the interaction of TLRs, among others [51]. Although we did not evaluate TLR expression, CD56^bright^CD16^+^ NK cells might have higher expression of TLR-3, TLR-7 and/or TLR-9, which could have favored IFN-γ production upon in vitro stimulation with BCG-CFP. The absence of neutrophil activation according to surface and intracellular markers and specific Th1 responses in this group of individuals might indicate that this response to BCG does not occur or that the time to evaluate such a response is not appropriate, because the cells evaluated in this group were immature. This heterogenic response to BCG was also observed by Oliveira et al. [52], who evaluated the immune response to BCG in a Brazilian clinical trial and showed that, two months after BCG vaccination, the CD4 T-cell-specific responses varied greatly among the participants of the study.

A rapid early innate and adaptive response to a vaccine should be a critical hallmark of protection. In this sense, our results showed that, contrary to the prediction, individuals whose NK cells did not respond by producing IFN-γ upon vaccination presented activation of neutrophils and early specific CD4 T-cell responses with a broad cytokine profile. One possible explanation for this is that cytotoxic NK cell activation and the increased expression of surface molecules and arginase by neutrophils could be important for the activation of macrophages to produce IL-12 and culminate in the induction of Th1-specific responses, among others, observed here in group 1 individuals [53].

We observed that the vaccine did not stimulate a significant increase in the number of neutrophils or in the expression of TNF-α and arginase in the groups analyzed on days 1 and 15. However, arginase was more highly expressed in groups 1 and 3 than in group 2. Neutrophils are the first defense against bacteria and perform phagocytosis, degranulation and nuclear material release functions in the form of extracellular neutrophil traps (NETs) [54,55]. Arginase-1, released by neutrophil granules, degrades arginine by an immunoregulatory homeostatic mechanism [56], causing inhibition of cytokine production by NK cells. Thus, it is interesting to observe that NK cells from group 1 individuals who presented a cytotoxic profile had higher expression of arginase, which may play a role in the reduction of IFN-γ expression following BCG vaccination when compared to the individuals with proinflammatory profile NK cells, who had higher expression of IFN-γ and low levels of arginase (group 2).

In this study, cryopreserved whole blood and PBMC samples were collected before and 15 days after BCG vaccination; thus, we were unable to predict the adaptative immune response to *M. tuberculosis* Ag85 after this time. Neutrophils were evaluated without in vitro stimulation to avoid degranulation and cell death, while all other cells were evaluated in vitro after stimulation with BCG-CFP or Ag85. At this point evaluated here, innate immunity is established and has already initiated the activation events of adaptive immunity, but it is not possible to predict the subsequent events of this response. Studies with longer periods of immune response evaluation following BCG Moscow vaccination are needed to confirm if the differences observed here are maintained.

## 5. Conclusions

The initial decreased expression of CD314 by NK cells correlated with the ability of NK cells to produce IFN-γ in response to BCG vaccination. However, the expression of CD314 is not a sine qua non condition for NK subpopulations’ cells to produce IFN-γ. Individuals who showed an increase in NK cell IFN-γ responses did not exhibit enhanced neutrophil activation or the same specific T-cell response heterogeneity as in other groups. Thus, the NK cell IFN-γ response, generation of NK subpopulations and the specific T immune response at 15 days after vaccination varied according to the initial CD314 expression by overall NK cells.

## Figures and Tables

**Figure 1 vaccines-11-01297-f001:**
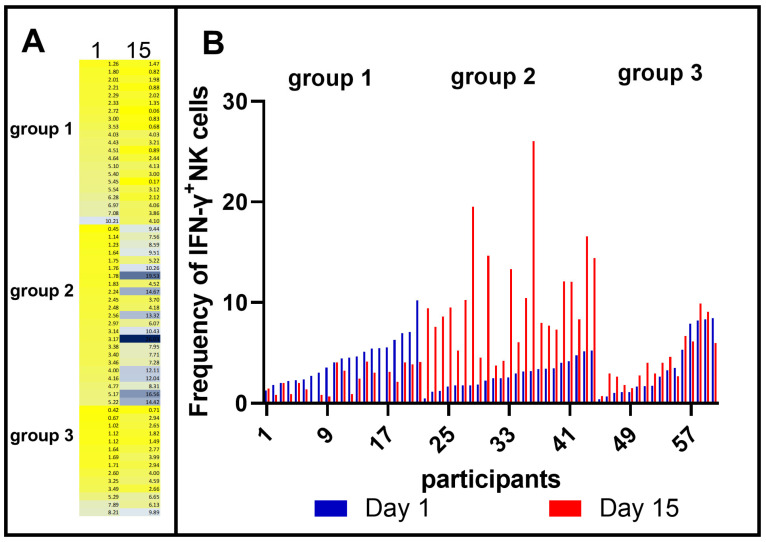
IFN-γ^+^ NK cells of individuals on day 1 and day 15 after BCG Moscow vaccination were classified according to IFN-γ^+^ percentage variation. In (**A**), the IFN-γ^+^ NK cells were classified according to the response magnitude, with color grading ranging from yellow (for the smallest amount) to blue (for the highest amount). Individuals who had a decrease in IFN-γ^+^ NK cell percentage (color of column 15 is lighter than column 1) were assigned to group 1. Individuals with increased IFN-γ^+^ NK cell percentage at day 15 (color of column 15 is darker than column 1) comprised group 2. Group 3 had the same IFN-γ^+^ NK cell frequencies for both times. In (**B**), the data from the grouped individuals were plotted as a bar graph, where, for each participant, the first bar shown in blue corresponds to day 1, and the red bar to the right represents paired IFN-γ^+^ percentage at 15 days postvaccination.

**Figure 2 vaccines-11-01297-f002:**
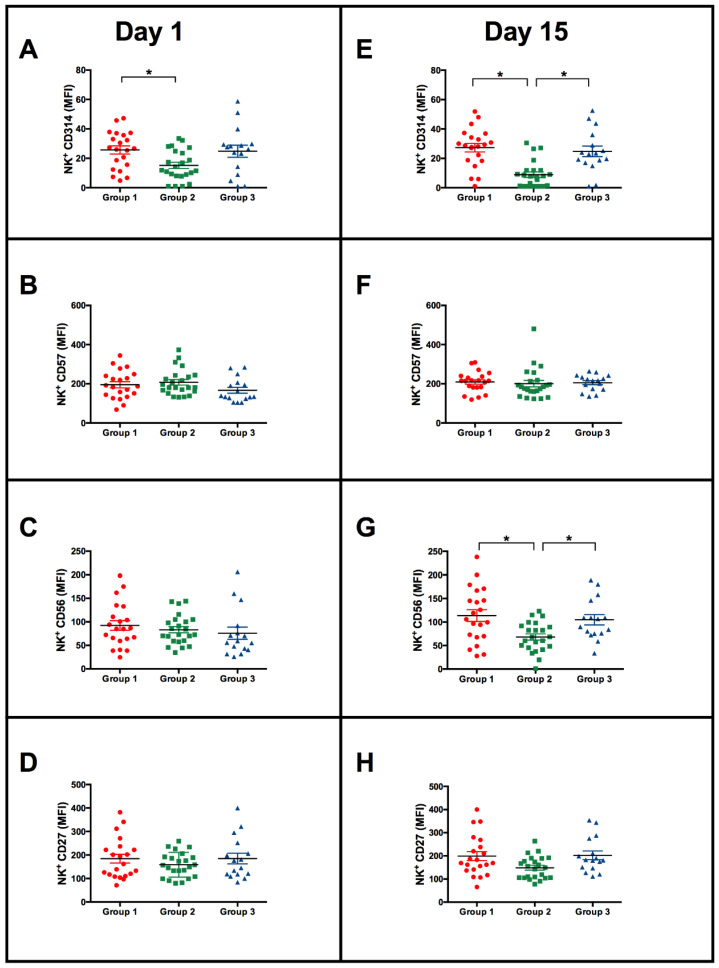
NK cell median fluorescence intensity (MFI) of CD314, CD57, CD56 and CD27 before and after BCG vaccination. NK cell CD314 (**A**), CD57 (**B**), CD56 (**C**) and CD27 (**D**) MFI of groups 1, 2 and 3 at baseline (day 1). In (**E**–**H**) MFI of NK CD314, CD57 CD56 and CD27 15 days post-BCG vaccination, respectively (Day 15). Significant differences after Kruskal–Wallis followed by Dunn’s multiple comparisons tests are shown with lines matching the pairs of groups. * *p* < 0.05.

**Figure 3 vaccines-11-01297-f003:**
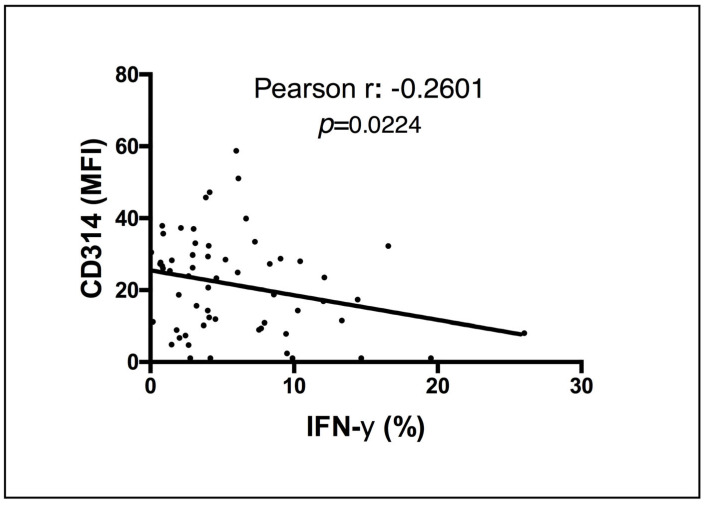
Correlation between the NK MFI of CD314 at baseline and the percentage of IFN-γ^+^ cells at 15 days post-BCG vaccination. Pearson test evaluation of the NK CD314 MFI at day 1 and percentage of IFN-γ+ cells at day 15. The *p* value is shown inside the graph.

**Figure 4 vaccines-11-01297-f004:**
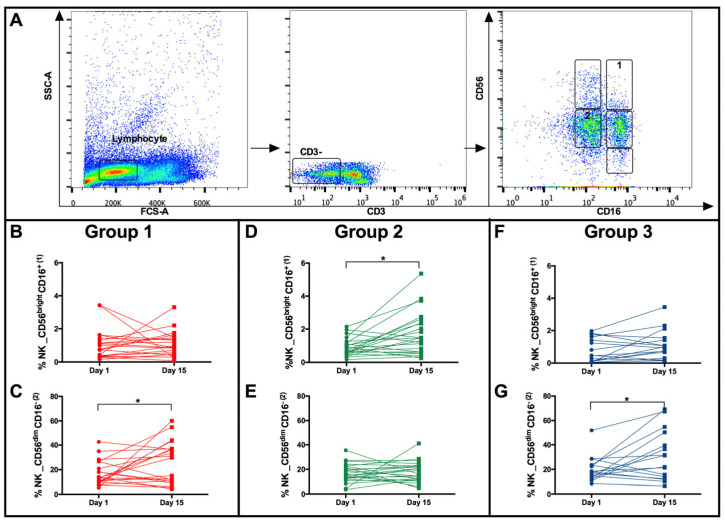
Subpopulations of NK cells at baseline and 15 days after BCG vaccination in individuals from groups 1, 2, and 3. (**A**) Gating strategy to evaluate NK subpopulations. CD3^−^ Lymphocytes were evaluated. Gate 1: CD56^bright^CD16^+^ and Gate 2: CD56^dim^CD16^−^. (**B**,**D**,**F**), percentages of the CD56^bright^CD16^+^ population in groups 1, 2, and 3, respectively. (**C**,**E**,**G**), percentages of the CD56^dim^CD16^−^ populations in groups 1, 2, and 3, respectively. Lines connect the percentages for each individual at baseline (Day 1) and 15 days postvaccination (Day 15). Statistical comparisons were made using paired *t* test, * *p* < 0.05.

**Figure 5 vaccines-11-01297-f005:**
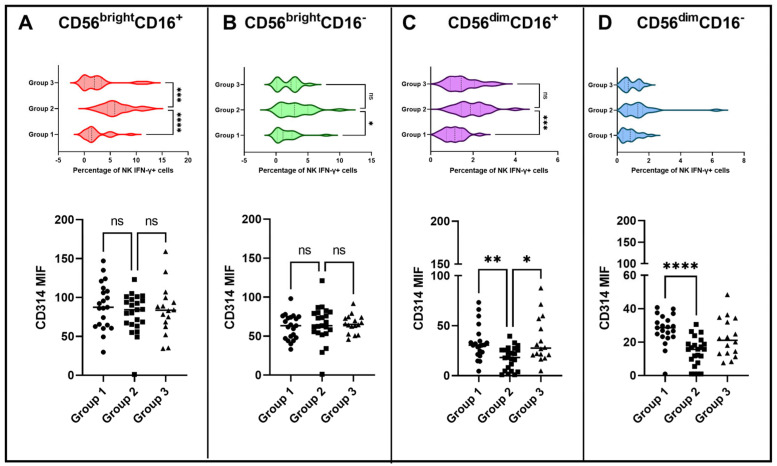
Subpopulations of NK cells at 15 days after BCG vaccination in individuals from groups 1, 2 and 3. Gating strategy to evaluate NK subpopulations is described in Appendix A. The CD56^bright^CD16^+^, CD56^bright^CD16^−^, CD56^dim^CD16^+^ and CD56^dim^CD16^−^ NK subpopulations were evaluated for the percentages of the IFN-γ^+^ and the CD314 MFI in (**A**–**D**), respectively. Top graphs show violin plots of the percentage distribution of IFN-γ^+^ NK subpopulations in groups 1, 2 and 3 (dark dotted lines represent the median and light dotted lines the interquartile range). Bottom graphs show MFIs of CD314 among different groups (horizontal lines represent the average of the MFIs). Significant differences after Kruskal–Wallis followed by Dunn’s multiple comparisons tests are shown with lines matching the pairs of groups. * *p* < 0.05; ** *p* < 0.01; *** *p* < 0.001; and **** *p* < 0.0001. ns—not significant.

**Figure 6 vaccines-11-01297-f006:**
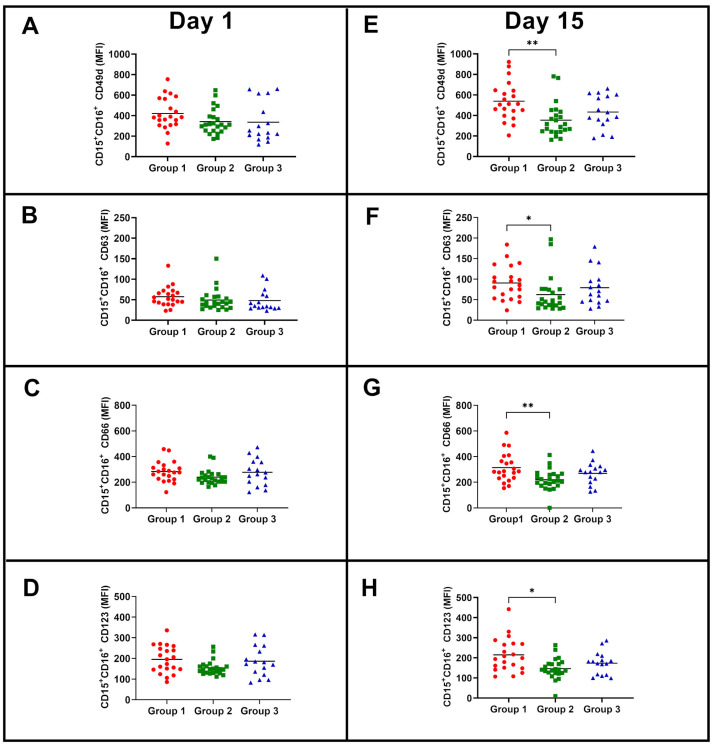
MFI of CD49d, CD63, CD66 and CD123 in CD15^+^CD16^+^ neutrophils among BCG-vaccinated participants. (**A**,**E**) show MFI of CD49d at days 1 (baseline) and 15 (postvaccination), respectively. (**B**,**F**) show CD63 MFI at days 1 and 15, respectively. (**C**,**G**) show CD66 MFI at days 1 and 15, respectively. (**D**,**H**) show CD123 MFI at days 1 and 15, respectively. The Kruskal–Wallis test with Dunn’s multiple comparisons was used to compare groups. * *p* < 0.05; ** *p* < 0.01.

**Figure 7 vaccines-11-01297-f007:**
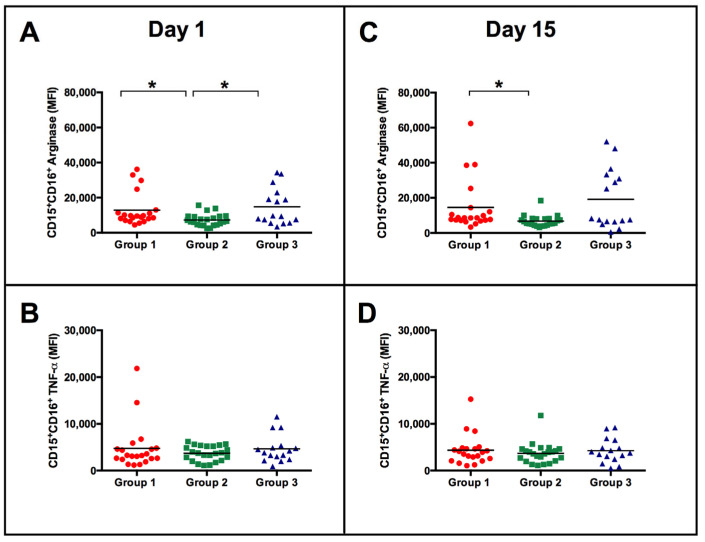
MFI of arginase and TNF-ɑ in CD15^+^CD16^+^ neutrophils among BCG-vaccinated participants. (**A**,**C**) show MFI of arginase on days 1 and 15 (post vaccination), respectively. (**B**,**D**) show TNF-ɑ MFI as described above. The Kruskal–Wallis test with Dunn’s multiple comparisons was used to compare the groups. * *p* < 0.05.

**Figure 8 vaccines-11-01297-f008:**
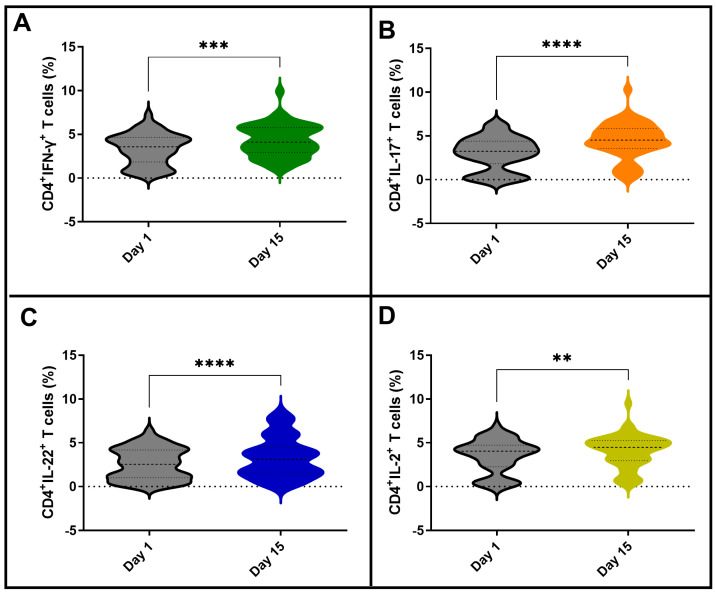
CD4^+^ T-cells expressing cytokines IFN-γ, IL-17, IL-22 and IL-2 in response to in vitro stimulation with Ag85 from *M. tuberculosis*. Frequency (%) of CD4^+^ T-cells expressing cytokines (**A**) IFN-γ, (**B**) IL-17, (**C**) IL-22 and (**D**) IL-2 at baseline (Day 1) and 15 days post-BCG vaccination (Day 15). The Wilcoxon test was used to compare Day 1 versus Day 15 analysis. *p* values: ** < 0.01; *** < 0.001; **** < 0.0001.

**Figure 9 vaccines-11-01297-f009:**
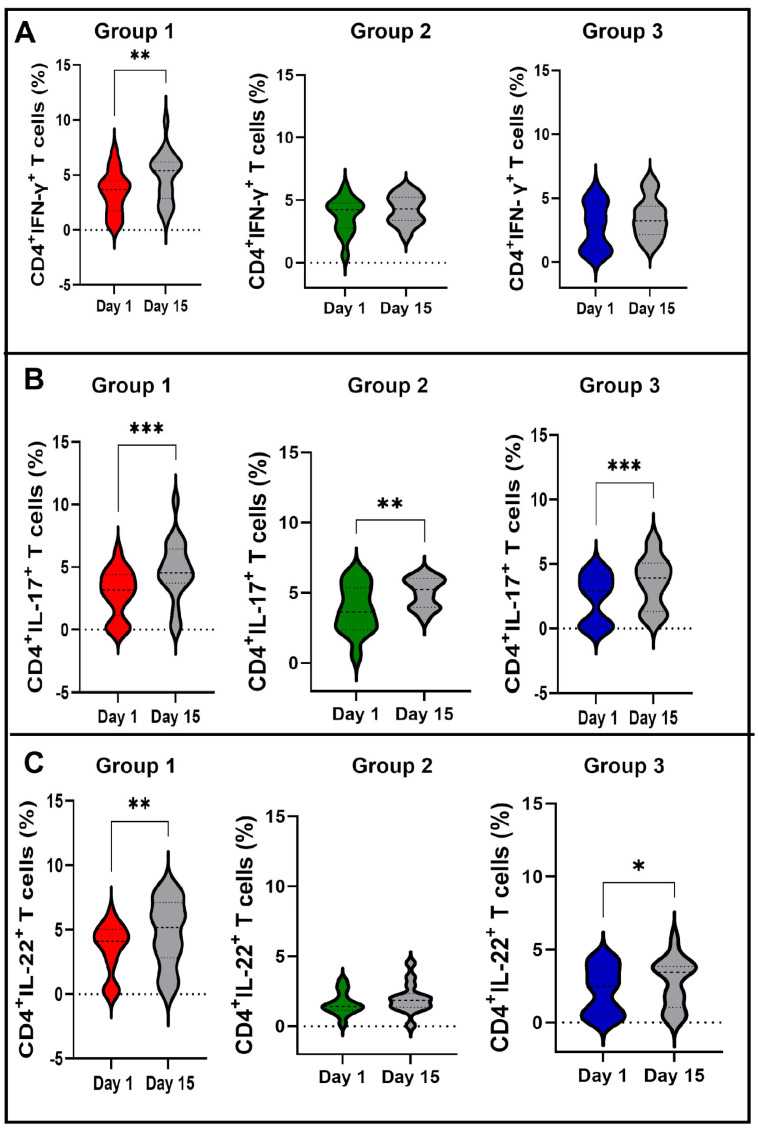
CD4^+^ T-cells expressing cytokines IFN-γ, IL-17 and IL-22 in response to recombinant Ag85 from *M. tuberculosis* according to the NK-IFN-γ MFI response to BCG vaccination. Frequency (%) of CD4^+^ T-cells expressing (**A**) IFN-γ, (**B**) IL-17, (**C**) IL-22 at baseline (Day 1) and 15 days post-BCG vaccination (Day 15) are shown. Red, green, and blue violins represent the results from day one of group 1, 2 and 3, respectively. Gray violins represent day 15 of all groups. The Wilcoxon test was used to compare Day 1 versus Day 15 analysis. *p* values: * < 0.05; ** < 0.01; *** < 0.001.

## Data Availability

Data supporting reported results are available upon request.

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
