# Peer review of "Decreased Expression of CD314 by NK Cells Correlates with Their Ability to Respond by Producing IFN-γ after BCG Moscow Vaccination and Is Associated with Distinct Early Immune Responses"

_vaccines, 2023, doi:10.3390/vaccines11081297_

Round 1

Reviewer 1 Report

The manuscript investigated if NK cell phenotype correlates to BCG's ability to activate the immune response. According to IFNg production of NK cells before and after BCG vaccination, the donors were divided into 3 groups. Then the authors checked several surface markers expressed on NK cells and found CD314 negatively correlated to IFNg level. Furthermore, they identified different NK subpopulations in 3 groups. Then the authors did a more comprehensive analysis of the samples. They checked neutrophils and CD4+ T cells. Overall, the findings are very interesting. However, I have one major concern as shown below.

The authors found that CD314 and CD56 expression is less in Group 2 (Fig. 2A, E). CD314 expression is negatively correlated with IFNg production. Further, the authors checked different NK subpopulations and found more CD56brightCD16+ NK cells in Group2 and more CD56dimCD16- NK cells in Group 1 and 3 after vaccination. All of those findings are interesting. However, there is no true connection between CD314, IFNg, and NK subpopulations. 

1) Did you check if CD314- NK cells can make more IFNg using FACS?

2) What is CD314 expression on different NK subpopulations?

3) What is the ability of IFNg production on different NK subpopulations? 

Author Response

RESPONSE LETTER

 REVIEWER #1:

The manuscript investigated if NK cell phenotype correlates to BCG's ability to activate the immune response. According to IFNg production of NK cells before and after BCG vaccination, the donors were divided into 3 groups. Then the authors checked several surface markers expressed on NK cells and found CD314 negatively correlated to IFNg level. Furthermore, they identified different NK subpopulations in 3 groups. Then the authors did a more comprehensive analysis of the samples. They checked neutrophils and CD4+ T cells. Overall, the findings are very interesting. However, I have one major concern as shown below.

The authors found that CD314 and CD56 expression is less in Group 2 (Fig. 2A, E). CD314 expression is negatively correlated with IFNg production. Further, the authors checked different NK subpopulations and found more CD56brightCD16+ NK cells in Group2 and more CD56dimCD16- NK cells in Group 1 and 3 after vaccination. All of those findings are interesting. However, there is no true connection between CD314, IFNg, and NK subpopulations.

Response:

The authors thank the reviewer 1 for carefully reading our manuscript. Here, we addressed if there was correlation between NK CD314 expression before BCG vaccination and the outcome of IFNg response to BCG observed at 15 days post vaccination.

Our hypothesis was that the initial NK phenotype could have guided the NK in vitro response to BCG. We have made this hypothesis more clear at the discussion section and in the revised title..

To clarify that we changed the conclusions accordingly.

Thank you.

1) Did you check if CD314- NK cells can make more IFNg using FACS?

Thank you for your question. Again, our hypothesis was to relate if the initial expression of CD314 was important for the ability to respond to BCG by producing IFNg after 15 days. We do not know if NK cells will produce IFNg based on the presence of CD314. In order to do that, it is necessary to sort these cells, stimulate and compare them in vitro. The NK cells evaluated in this clinical trial all expressed CD314, at different levels. Additionally, NK cells CD314- is a rare event, thus even if we did this analyses it would not have sufficient statistical power. Although, this could be an interesting point to be addressed in the future. Thank you.

2) What is CD314 expression on different NK subpopulations?

3) What is the ability of IFNg production on different NK subpopulations?

Thank you for queries 2 and 3. Now we have addressed them, and it allowed us to improve our manuscript.

The NK response to BCG vaccination was evaluated after stimulation with M. bovis CFP and is presented as percentage of IFNg positive cells, while the CD314 expression is shown by median intensity of fluorescence. All the results in the new figure 5 of the revised manuscript are from cultured NK cells obtained 15 days after BCG vaccination. The revised manuscript results section are in lines 253-262 as:

To evaluate which NK subpopulation contributed to the IFN-γ response after BCG vaccination, the percentage of IFN-γ+ NK cells was quantified (Figure 5). The percentage of NK CD56brightCD16+, CD56brightCD16- and CD56dimCD16+ subpopulations positive for IFN-γ was increased only in group 2. Possibly, these NK subpopulations contributed to the overall BCG IFN-γ response 15 days post-vaccination. Because it was observed that before vaccination the expression of CD314 was negatively associated with the IFN-γ response to BCG, the CD314 expression was also evaluated for each NK subpopulation. NK CD56brightCD16-, and CD56brightCD16+ cells presented similar MFI of CD314 among all groups, while NK CD56dimCD16-, and CD56dimCD16+ cells from group 2 presented reduced CD314 expression compared to the other groups.

And at the Discussion section lines: 371-395

Group 1 presented reduced production of IFN-γ by NK cells 15 days after vaccination (Figure 1). Such NK cells had increased CD314 expression compared to group 2 at baseline (Figure 2 A). In addition, both the CD314 and the CD56 molecules were increased 15 days after vaccination (Figure 2E and G) compared to the other groups. Interestingly, NK cells from individuals in group 3 that did not change their ability to produce IFN-γ upon BCG vaccination and in vitro stimulation also presented higher levels of CD314 and CD56 than NK cells from individuals in group 2. Apparently, the expression of CD314 in different stages of NK cell maturation alters the production of IFN-γ by NK cells after vaccination (Figure 3). Our results suggest that it is possible that high levels of CD314 expression may inhibit their ability to produce IFN-γ after vaccination and in vitro stimulation with BCG cultured filtrate proteins (BCG-CFP). Conversely, low levels of CD314 expression by NK cells may result in a higher capacity to produce IFN-γ after vaccination and in vitro stimulation with CFP. The in vitro stimulation with BCG-CFP showed that after BCG vaccination, CD56dimCD16+ and CD56dimCD16- NK subpopulations (Fig 4 A and Fig 5 C and D) from group 2 subjects presented reduced levels of CD314, thus were the main contributors for the reduced overall NK CD314 expression (Fig 2 E) compared to the other groups. The ensuing question would be that low expression of CD314 by NK subpopulations would be associated to a high positivity for IFN-γ. That was not the case, as the NK subpopulations CD56brightCD16+ and CD56brightCD16- were the main groups positive for IFN-γ and at 15 days post vaccination, they did not present differences in CD314 expression when compared amid the different groups. Therefore, the lower expression of CD314 is probably a phenotype associated with the initial status that may be directly or indirectly related to the IFN-γ responses to BCG, but it is not a phenotype associated with higher expression of IFN-γ and further studies should be done to understand the mechanisms involved in this characteristic.

Finally, the authors thank the reviewer one for the important queries that helped us to clarify the assumptions that we proposed in the manuscript.

Reviewer 2 Report

Recent studies demonstrated an association of BCG vaccination with a reduced risk of non-mycobacterial infections, allergies, cancer and overall mortality. These non-specific effects of BCG vaccination are due to cells of the innate immune system rather than specific memory T cells. The article under review is aimed at evaluation of the NK cells role in formation of BCG-induced trained immunity.

My concerns with the article are as follows:

1. The probands should be described in more details. It is not enough just to refer to the clinical trials design. One would like to know the age of probands, BCG vaccination history, TB contacts, etc. Such information might have a significant impact on the assessment of the correctness of the formation of probands groups.

2. BCG vaccine dose should be stated.

3. The gating strategy for the immunofluorescent stainings should be added as Supplement.

4. The only conclusion made by the authors as a result of their investigation is that "the NK cell IFN-γ response to BCG vaccination impacts the early immune response". Does one really need to take that much of trouble to make such a general conclusion?

Overall quality of English language is acceptable, minor editing is required.

Author Response

REVIEWER #2:

Recent studies demonstrated an association of BCG vaccination with a reduced risk of non-mycobacterial infections, allergies, cancer and overall mortality. These non-specific effects of BCG vaccination are due to cells of the innate immune system rather than specific memory T cells. The article under review is aimed at evaluation of the NK cells role in formation of BCG-induced trained immunity.

My concerns with the article are as follows:

  1. The probands should be described in more details. It is not enough just to refer to the clinical trials design. One would like to know the age of probands, BCG vaccination history, TB contacts, etc. Such information might have a significant impact on the assessment of the correctness of the formation of probands groups.
  2. BCG vaccine dose should be stated

Response:

The authors thank the reviewer 2 for carefully reading our manuscript and proposing important changes to improve it.

 For the overall demographics of the enrolled subjects of this clinical trial, we added the reference Dos Anjos et al. (2022), [Dos Anjos LRB et al. (2022) Efficacy and Safety of BCG Revaccination With M. bovis BCG Moscow to Prevent COVID-19 Infection in Health Care Workers: A Randomized Phase II Clinical Trial. Front. Immunol. 13:841868. doi: 10.3389/fimmu.2022.841868].

 To clarify the demographic characteristics of the analysed participants in the present work, after grouping them according to the NK IFN BCG responses we added one paragraph in the methods sections: lines 105-116:

Heparinized total blood was drawn from the vaccinated participants on the day of inclusion (day 1) and 15 days after randomization. Briefly, after participant clinical trial inclusion, the blood was drawn, and the participant received a sealed envelope containing their group assignment, “vaccination” or “no vaccination” with BCG [29]. Individuals that were randomly assigned to the “vaccination” group were intradermally vaccinated with 100μL (106 CFU) of the BCG Moscow vaccine in the deltoid region of the right arm by an experienced nurse. In this study, 60 individuals who received the BCG Moscow vaccine were evaluated. For the comparison of NK cells producing IFN-γ at days one and 15, 48 individuals from the “no vaccination” group were also analysed. The median age of the participants were 42 years old, ranging from 18 to 60 years old, most of them were female (68%). All participants had previous BCG scars and were exposed to several infectious diseases including tuberculosis, once they worked at the General Hospital.

  1. The gating strategy for the immunofluorescent stainings should be added as Supplement.

Response:

The gate strategies were added to the supplementary materials.

  1. The only conclusion made by the authors as a result of their investigation is that "the NK cell IFN-γ response to BCG vaccination impacts the early immune response". Does one really need to take that much of trouble to make such a general conclusion?

Response:

Thank you for your comments and apologize if we were not clear enough about our findings. We have revised such statements in the abstract (unfortunately because of space limitations we could not elaborate more) and in the conclusion section, thank you. They now read as follows:

For the abstract:

In conclusion, the expression of CD314 by NK cells before BCG vaccination influences their IFN-γ responses, generation of NK subpopulations, and the specific T immune response at 15 days after vaccination.

For the conclusion section:

The initial decreased expression of CD314 by NK cells correlated with the ability of NK cells to produce IFN-γ in response to BCG vaccination. However, the expression of CD314 is not sine qua non condition for NK subpopulations cells to produce IFN-γ. Individuals who showed an increase in NK cell IFN-γ responses did not exhibit enhanced neutrophil activation or the same specific T-cell response heterogeneity as in other groups. Thus, the NK cell IFN-γ response, generation of NK subpopulations and the specific T immune response at 15 days after vaccination varied according to the initial CD314 expression by overall NK cells.

Round 2

Reviewer 2 Report

After the corrections made the article might be recommended for publication.